# Uncertainty of Quantitative X-ray Fluorescence Micro-Analysis of Metallic Artifacts Caused by Their Curved Shapes

**DOI:** 10.3390/ma16031133

**Published:** 2023-01-28

**Authors:** Tomáš Trojek, Darina Trojková

**Affiliations:** Department of Dosimetry and Application of Ionizing Radiation, Faculty of Nuclear Sciences and Physical Engineering, Czech Technical University in Prague, Břehová 7, 11519 Prague, Czech Republic

**Keywords:** X-ray fluorescence, quantitative analysis, metal, Monte Carlo method

## Abstract

This paper summarizes the effects of irregular shape on the results of a quantitative X-ray fluorescence (XRF) micro-analysis. These effects become relevant when an XRF analysis is performed directly on an investigated material. A typical example is XRF analyses of valuable and historical objects whose measurements should be performed non-destructively and non-invasively, without taking samples. Several measurements and computer simulations were performed for selected metallic materials and shapes to evaluate the accuracy and precision of XRF. The described experiments and the corresponding Monte Carlo simulations were related to the XRF device designed and utilized at the Czech Technical University. It was found that the relative uncertainty was typically about 5–10% or even higher in quantitative analyses of minor elements due to irregular shapes of surfaces. This must be considered in cases of the interpretation of XRF results, especially in the cultural heritage sciences. The conclusions also contain several recommendations on how to measure objects under hard-to-define geometric conditions with respect to reduction in the surface effect in quantitative or semi-quantitative XRF analyses.

## 1. Introduction

Nowadays, various methods of chemical and structural analysis are used to investigate cultural heritage objects. One of them is an analytical technique known as X-ray fluorescence (XRF) analysis, which is based on the stimulated emission and spectroscopy of so-called characteristic X-rays. When a material is exposed to ionizing radiation (X-rays, in this particular case), the absorption of photons causes the excitation and ionization of atoms. The process of de-excitation can be followed by the emission of characteristic X-rays. The energy of characteristic X-rays exactly corresponds to the energy difference between the excited and ground states of inner-shell electrons, and this value is particular for each shell transition and element. Individual chemical elements can be identified and even quantified because the intensity of characteristic X-rays is related to the concentration of an element in a material.

XRF analyses have been used for a long time as a classical laboratory analytical method and involve taking a sample from a material, preparation to the desired form, and measurement under well-defined experimental conditions. Thanks to the significant progress in X-ray instrumentation in the last 25 years, XRF has become a method suitable for in situ analysis [1], and laboratory measurements have begun to be carried out more often directly on investigated objects without the necessity of taking any samples. Thus, XRF analysis can be performed non-destructively and non-invasively. This was the turning point for its application in cultural heritage sciences [2,3]. The most frequently analyzed materials are inorganic pigments in paintings [4], ceramics [5], glass [6], and metals [7] because almost all chemical elements with an atomic number of 13 (aluminum) and higher can be identified with XRF. Even natrium (Z = 11) can be identified if measurements are performed in a vacuum chamber or in a helium atmosphere. The structure of art and archeological objects can be quite complicated, and materials of different compositions are often located very close to each other. In these cases, an XRF micro-analysis, which is based on the focusing of an excitation X-ray beam on a small spot with X-ray optics [8], must be applied.

The standard procedure for a quantitative XRF analysis includes measurements of reference materials of known compositions and the calibration of an XRF setup for certain experimental conditions. When the relations of X-ray intensities (detected count rates) versus elemental concentrations are drawn, they can be used for the determination of elemental concentrations in unknown samples. It is presumed that unknown samples and reference materials are analyzed under the same experimental conditions and have similar matrices. 

If XRF analyses are performed directly on art and archeological objects, their surfaces are usually not flat and smooth, and thus, the results of the quantitative analysis are affected by these surface effects [9,10,11]. Since a collimated or focused X-ray beam is narrow and the mean free path of low-energy X-rays in materials is approximately tens of micrometers, the effect of surface irregularities on X-ray transport in a material is almost equivalent to the local incline of a flat surface in a certain direction. Irregular shapes result in a decrease or an increase in X-ray intensities. If the primary X-rays are absorbed in a material closer to the surface and the characteristic X-ray path in a material toward a detector is shorter, then the X-ray absorption in the material is lower, and the detected intensities of characteristic X-rays are higher [12]. Such an effect is more significant for a geometric arrangement with a large angle between the excitation and detected X-ray beams [13]. 

Metals represent one of the most frequently analyzed groups of materials because almost all elements present in historical metallic materials can be identified with XRF. It is usually considered that the sum of the concentrations of all the measured elements in a metal is equal to 100%. This is especially important in the case of XRF analyses affected by irregular surface. If the characteristic X-ray intensities of all the present elements are affected qualitatively and quantitatively in the same way, the elemental concentration ratios remain constant, and even absolute elemental concentrations are not modified due to surface effects because normalization to 100% can be applied. However, characteristic X-rays have different energies for various elements, and therefore, irregular shape has quantitatively different effects on individual elements or their particular X-ray lines. Thus, the accuracy and precision of XRF analyses are also affected.

The main goals of this research are the study of surface effects in XRF, as well as pointing out some pitfalls in the quantitative analysis of uneven surfaces and estimation of the uncertainty of quantitative analyses for some historical materials.

## 2. Materials and Methods

### 2.1. Instrumentation

The experiments and theoretical calculations described in this paper are related to a tabletop device for XRF micro-analysis designed and used to analyze various historical artifacts at the Czech Technical University in Prague (Prague, Czech Republic). Its photograph is shown in Figure 1. This XRF system consists of an SDD detector (Amptek Inc., Bedford, MA, USA), an air-cooled X-ray tube with a Mo anode and a maximum power of 50 W (50 kV, 1 mA), and a positioning system for XRF scanning. The X-ray tube includes a fixed polycapillary focusing optics device. According to the manufacturer of the optics device, it has a 15 μm focal spot (FWHM for 17.4 keV) at a working distance of 4 mm from the output end of the optics device. The X-ray spectrum of the X-ray tube is modified by the attached polycapillary focusing optics device (X-ray Optical Systems Inc., East Greenbush, NY, USA), and its expected shape was described in a previous publication [12]. The X-ray beam of the X-ray tube strikes perpendicularly the surface of an analyzed object, and the produced radiation is detected at an angle of approximately 45° with an Amptek SDD detector (25 mm^2^ × 0.5 mm, 1/2 mil Be window). The entrance window of the detector is placed at a distance of 1 cm from the analyzed point located on the surface of an object. The procedure for the analytical approximation of the X-ray tube spectrum was described in detail in a previous article [14].

### 2.2. Monte Carlo Simulation

The Monte Carlo method is a suitable numerical calculation method for math problems that are difficult to solve analytically. It is based on generating random or, rather, pseudo-random numbers and simulating radiation transport in matter. Common tasks of quantitative XRF analysis can be solved analytically using equations derived from Sherman’s equation [15]. If the studied problem is more complicated and considers irregular shapes, for instance, the application of the Monte Carlo method is recommended. The Monte Carlo code MCNP (Monte Carlo N-Particle Transport Code (Los Alamos National Laboratory, Los Alamos, NM, USA) running on a normal PC has been used for recent calculations. It provides the intensities of characteristic X-rays produced in an object of a selected shape and composition. X-ray fluence spectra in the position of the detector are scored with the MCNP code using Tally F5. The application and validation of this code in X-ray fluorescence analyses have been already described in previous publications [16,17,18]. The computing time is approximately five minutes for an individual task.

## 3. Results and Discussion

### 3.1. XRF Scanning of a Modern Steel Ball Bearing

The surface effects in XRF could be easily demonstrated in an example of a modern steel ball bearing. This spherical object with a diameter of 5 mm had a smooth but curved surface. Apart from the major matrix element (iron), it contained 1.5% chromium and about 0.3% manganese. It was assumed that these elements were distributed homogeneously in the whole volume of the ball bearing. The aim of the experiment was to show how the intensities of characteristic X-rays of iron and chromium (Kα lines) depended on the position illuminated with the X-ray microbeam. The characteristic X-rays of manganese were omitted from this investigation because of the lower concentration and the interference of the Mn-Kα line with the line of chromium Cr-Kβ. The ball bearing was scanned with the micro-XRF system in both horizontal directions (X, Y) with a step of 0.1 mm. The acquisition time was 2 s per one spot. The X-ray tube current was set to only 0.2 mA because of the high efficiency of the production and detection of characteristic X-rays for both these elements. Since the diameter of the sphere was 5 mm, a scanning area of 5.6 mm × 5.6 mm (56 × 56 = 3136 spots) was selected to cover the whole ball bearing. The sum spectrum, i.e., the sum of these 3136 individual spectra, included the peaks of chromium, manganese, and iron (see Figure 2).

Figure 3 shows significant differences in the number of detected photons (net peak areas in individual X-ray spectra) of Cr-Kα and Fe-Kα lines depending on the position of the excitation X-ray beam. The central parts of both images correspond to the measurements of the top of the ball where the direction of the incident beam was perpendicular to the surface plane. These data would be like the results for a flat object in reference geometry. Since the detector was oriented in the direction of the x-axis at the bottom side of the images, higher X-ray intensities were acquired when the X-ray beam irradiated the parts of the sphere inclined toward the detector. This result was reasonable because X-rays produced at a certain depth must penetrate only a short distance into the steel and, therefore, are less absorbed in a material. Much lower X-ray intensities were obtained in the upper parts of the images, where the farther side (dark side) of the ball was measured and the produced X-rays must penetrate a long distance through the steel material toward the detector. The images of the sphere in Figure 3 are not circular but are deformed, especially on the dark side where high X-ray attenuation prevented the XRF analysis. The characteristic X-ray images of the ball bearing were slightly turned counter-clockwise because the detector was not placed exactly in the direction of the scanning x-axis.

This extreme variation in X-ray intensities depending on the position of analysis (local orientation of the surface) disabled our ability to calculate the concentrations of the elements directly from the values of detected photons. On the other hand, it seemed that a quantitative or semi-quantitative XRF analysis was possible using the Cr-Kα/Fe-Kα ratio. We could use the advantage that such a surface effect was qualitatively similar for both elements (increased intensity on the inclined side and decreased intensity on the dark side). However, it seemed that the Cr-Kα/Fe-Kα ratio was not constant, and the determined concentrations of elements could differ from the correct ones. Figure 4 shows the concentrations of Cr in the steel determined along the x-axis (direction towards the detector). The dots represent results from the XRF scanning when the Cr-Kα/Fe-Kα ratio was recalculated to chromium concentration using a calibration curve (Cr concentration versus Cr-Kα/Fe-Kα ratio for the flat surface). The line shows the results of the Monte Carlo simulations fitting the real measurements. A higher concentration of chromium was obtained on the inclined side and a lower concentration on the dark side with respect to the expected value of 1.5%. Figure 4 also shows good correlation of the experiment with the Monte Carlo simulations, and thus, this result can also be considered as validation for the used Monte Carlo model. Further investigations of surface effects in the XRF micro-analysis were based on the calculations of X-ray intensities for various materials and shapes with the Monte Carlo code MCNP.

### 3.2. Monte Carlo Simulation of XRF Micro-Analysis of Sterling Silver Spherical Objects

There are a lot of historic metallic materials that were used for artifact production of various shapes. For instance, sterling silver was used for currency and general goods. Sterling silver is an alloy of silver containing 92.5% silver by weight and 7.5% other metals by weight, usually copper. We decided to perform initial calculations just for a silver alloy containing 92.5% silver and 7.5% copper. Such material is suitable for studying surface effects because the spectrum of characteristic X-rays includes low-energy L-lines of silver (~3 keV), high-energy of Kα line of silver (25.3 keV), and a Kα line of copper located approximately in the middle of the X-ray spectrum (8 keV). 

The first simulation of X-ray fluorescence scanning was performed for a sphere with a radius of 1 mm, and the step of this virtual scanning was again 0.1 mm. A simplified scheme of the geometric arrangement is shown in Figure 5, where the excitation X-ray beam of the X-ray tube fell on the top of the sphere at the XYZ position: (0, 0, 1). The source and the detector were moved together in the XY plane to simulate real XRF scanning. The X-ray intensities of individual X-ray lines were scored for each position. The X-ray intensities were then recalculated to elemental concentrations and drawn. If the X-ray intensities were 10 times lower than expected for a flat sample in reference geometry, the elemental concentrations were not evaluated. This was the same as in real XRF analyses, when X-ray spectra with suspiciously low numbers of counts are discarded because they are evidently measured incorrectly. The first concentration map shown in Figure 6 indicates the values of copper concentrations obtained in the XRF micro-analysis by illuminating different positions on the surface of the sphere. The red circle in this figure shows the border of the sphere. The concentration of silver was a supplement of 100%. In this case, X-ray intensities of Cu-Kα were compared with the intensity of Ag-L, i.e., a calibration diagram of copper concentration versus the Cu-Kα/Ag-L ratio was used. Similar calculations of copper concentration were performed using the Ag-Kα line (calibration based on Cu-Kα/Ag-Kα ratio), as shown in Figure 7. Both these figures demonstrate an overestimation of the copper concentration when the X-ray beam of the source illuminated the surface inclined toward the detector. Concentrations of copper lower than 7.5% were determined on the dark side of the sphere, or they were not evaluated at all because of very low X-ray intensities. It seems that the evaluation of concentrations using Ag-L lines suffered from lower fluctuations, and the elemental concentrations were determined more precisely using the L-lines. Such results are discussed later. 

The same calculations were performed for a silver sterling sphere with a 3 mm radius. Similar results demonstrated that such surface effects do not depend significantly on the dimensions of an investigated object (or surface irregularity) if it is much larger than the mean free path of X-rays in a material.

### 3.3. Monte Carlo Simulation of XRF Micro-Analysis of a Sphere-Shaped Hole in Sterling Silver

Another object was represented by a sphere-shaped hole in sterling silver, as shown in Figure 8. Concentration maps of copper are displayed in Figure 9 and Figure 10 representing the results obtained with the Ag-L and Ag-Kα lines, respectively. In these cases, X-rays were detected only from the left side of the object. Since the surface was inclined toward the detector in this measurable area of the object (left side in Figure 9 and Figure 10), the concentration of copper was overestimated again. Similarly, worse results (higher deviation from 7.5%) were determined when the Ag-Kα line was used in quantitative evaluation instead of the Ag-L line. 

When the right side of the sphere-shaped hole (x > 0) was analyzed, the produced characteristic X-rays must penetrate a long distance in the sterling silver material, and the strong X-ray absorption prevented the evaluation of concentrations in practice. Therefore, we applied the same restriction to not evaluate concentrations if the X-ray intensities were 10 times lower than expected for such material in standard geometry.

### 3.4. Monte Carlo Simulation of XRF Micro-Analysis of Flat Sterling Silver Coin with Two Sharp Edges

The results described above are related to the analysis of silver jewelry. When silver coins or medals are analyzed, disturbing surface effects can complicate or even disable quantitative evaluation. The surfaces of coins and medals are usually flat, but any irregularities can result in different absorptions of characteristic X-rays. A typical example is a relief represented by several small edges on a surface. We considered two sharp edges with a height of 0.05 mm on the surface of a sterling silver coin, as shown Figure 11. The effect of these edges on the X-ray intensities of Ag-L, Ag-Kα, and Cu-Kα lines was calculated again with the Monte Carlo simulations. The correct X-ray intensities corresponding to flat and smooth surfaces were obtained with calculations for the areas far from the edges. When the X-ray beam was focused on the edge on the right (position x = 1 mm = 1000 μm), characteristic radiation could be easily detected due to lower X-ray absorption toward the detector. The reason was the partially missing material for x > 1 mm. The question was whether the increase in X-ray intensities would be the same or similar for all three X-ray lines. The answer can be found in Figure 12, where the calculated X-ray intensities are drawn for various positions of the X-ray beam along the x-axis. Unfortunately, the intensity of Cu-Kα was much more increased close to the edge than the intensities of both silver lines. This overestimated the copper concentration again. The value of 100% in Figure 12 corresponds to the X-ray intensity of a certain element for the flat and smooth surfaces of the sterling silver coin. The effect on the Ag-Kα line was evident even several tens of micrometers from the edge because of the high energy and the lower attenuation coefficient of the Ag-Kα line. 

Then, the coin was simulated close to the second edge (position x = −1 mm = −1000 μm), where the presence of this additional 0.05 mm thick layer of silver caused very strong absorption, especially for X-rays with lower energy (Ag-L and Cu-Kα), as shown in Figure 13. If the area located just in front of the edge (x < −1 mm) was illuminated with the X-ray beam, the spectrum included almost only the X-rays of Ag-Kα, and the material seemed to be pure silver. However, technicians in X-ray laboratories should notice the absence of Ag-L lines and, thus, consider such experimental data risky for quantitative analysis. The dimensions of this shielded area depended on the height of the edge. It is evident that the presence of sharp edges on the surfaces of analyzed materials could significantly affect the X-ray intensities and, consequently, the elemental concentrations. On the other hand, this happens only if the excitation beam is focused on the surface very close to an edge. Therefore, the probability of this effect is quite small, and technicians can reduce it by selecting more suitable areas for analysis.

### 3.5. Uncertainty of Quantitative XRF

If it is not possible to analyze a flat and smooth surface for reference geometry and the local orientation of an analyzed surface is unknown, one must consider the uncertainties in elemental concentrations caused by these disturbing effects. The previously introduced virtual XRF micro-analyses of spherical objects shown in Figure 5, Figure 6, Figure 7, Figure 8, Figure 9 and Figure 10 can help us to estimate the range of concentrations the can be determined with an XRF micro-analysis of materials with generally unknown surface orientations. A sphere is a good object for such an investigation because different points of its surface represent all possible orientations of the surface of a real object. Therefore, we made dozens of Monte Carlo simulations for each material and geometry represented by spheres (or sphere-shaped holes in materials) with a radius of 1 mm. This virtual XRF scanning provided a set of elemental concentrations that could be obtained in the real XRF analysis of an object randomly oriented in space. This set of elemental concentrations could be described utilizing a range of concentrations (a minimum and a maximum value), a mean value, and a standard deviation. Such results could give an estimate of the uncertainty of elemental concentrations caused by curved or rough surfaces. All these results are summarized in Table 1. The concentrations were evaluated again only in those cases when the X-ray intensities were higher than 10% of the expected values for a flat object measured with reference geometry. 

The results (a–d) refer to previously described calculations for objects (spheres or sphere-shaped holes) made of sterling silver. The ranges of the obtained concentrations show that some results for copper concentrations were quite far from the correct value of 7.5%. However, the mean concentration of copper was almost correct for the spherical object because a higher concentration of copper from the inclined side was compensated for by a lower value on the dark side. In the case of the sphere-shaped hole, the mean concentrations were wrong because the X-ray intensities could be evaluated only from one part of the hole. The simulations also showed that elemental concentrations were determined more precisely using the L-lines of silver. The reason was the difference in attenuation coefficient for the Cu-Kα line and the corresponding Lα lines of silver. The Ag-Kα line had a very low attenuation coefficient due to its high energy, and therefore, it was quite far from the attenuation coefficient for the Cu-Kα line. Since the attenuation coefficients for the Cu-Kα and Ag-L lines were closer to each other, an irregular shape of the surface increased or decreased these two lines quite similarly, i.e., the intensity ratios of these two lines remained almost constant, and thus, the element concentrations were close to the correct values if normalization of concentrations (to 100%) was considered. 

Similar results were achieved for a silver alloy made with only 50% silver and 50% copper (e–f). In this case, quite a small variation in copper concentration seemed to be acceptable in the quantitative analysis of real silver objects. 

The remaining results (g–) show the variations in determined concentrations for three other historical materials: 18-carat gold (Au 75%, Ag 12.5%, and Cu 12.5%), bronze (Cu 79%, Sn 7%, and Pb 14%), and steel with only one minor element (Fe 95% and Mn 5%). Simulations were performed for the sphere again, and the L-lines of silver and tin were used for quantitative analysis. These conditions should be more suitable for a quantitative analysis. In the case of the 18-carat gold, the concentration of copper was determined more precisely than in silver because the Kα-line of copper was quite closer to the L-lines of the major element (gold). The worst result was the variation in Sn content in bronze, where the relative deviation of tin concentration (standard deviation/mean concentration) equaled 0.166, i.e., 16.6%. 

A very precise result was obtained for manganese in steel (i). Another important material, brass, is not included in Table 1. However, the determination of zinc in such a copper alloy should also be accurate because the Kα lines of copper and zinc have similar X-rays energies below their X-ray absorption edges, as analogously observed for manganese in steel.

### 3.6. Quantitative XRF Analysis of Tilted Objects

The XRF analysis of an object with a flat surface that is tilted from the reference plane is, in fact, comparable with the analysis of an object with a curved surface because the local orientation of the surface (with respect to the direction of an incident beam and the direction towards a detector) is always crucial. The Monte Carlo results displayed in Figure 6, Figure 7, Figure 9, and Figure 10 show almost the same elemental concentrations for the same position in the x-axis (similar concentrations in individual columns). This indicates that the tilting of a flat object in the x-axis should not have a significant effect on the X-ray intensity ratios and determined concentrations. 

Thus, we tried to prove this hypothesis experimentally. A piece of bronze with a flat surface was analyzed with our XRF device for micro-analysis several times at different geometries. Firstly, it was examined at a reference geometry where the object lay horizontally, and the angles were 90° for the incident beam and 45° for the emergent characteristic X-rays. Then, it was subsequently tilted 30 degrees in the x-axis, in the y-axis with inclination toward the detector, and finally in the y-axis in the opposite direction, as shown in Figure 14. The orientation of the axis was the same as in the previous figures. The tilting (30 degrees) always referred to the reference geometry. The main components of this copper-based alloy were copper, zinc, tin, and lead. Their concentrations determined under these four geometrical conditions are summarized in Table 2. When this piece of bronze was tilted in the x-axis and analyzed, the elemental concentrations were almost the same as those of the reference geometry in which the XRF device was calibrated, and the quantitative analysis should be correct. However, tilting of the object in the y-axis in any direction caused a change in the attenuation of individual X-ray lines, resulting in significant variation in the concentrations of tin and lead. The concentration of zinc was only slightly affected by this effect because its X-ray energy was close to the energy of the copper Kα line.

## 4. Conclusions and Recommendations

The surface effects in the X-ray fluorescence micro-analysis of metallic artifacts were quantified by applying the Monte Carlo calculation method. Its main advantage is a quite short computing time for XRF experiments and the possibility of simulating almost arbitrary geometric arrangements and any composition of materials. Attention was paid especially to historical metallic materials requiring non-destructive and non-invasive XRF micro-analysis. The presented results referred mainly to copper, silver, and gold alloys. Similar results could be obtained for iron alloys because iron (Z = 26) has similar X-ray properties as copper (Z = 29). The results of the Monte Carlo simulations presented in Table 1 show that the ratio of the standard deviation and the mean concentration was typically about 0.05–0.10 or higher for minor elements. This corresponded to a relative uncertainty of 5–10% or more in quantitative analyses of minor elements due to irregular shapes of surfaces. 

With respect to the presented results, we propose several recommendations for how to measure objects under hard-to-define geometric conditions. There are various materials and it is very difficult to draw general conclusions based on the results for only some of them. Nevertheless, the performed calculations revealed valuable information about the behavior of X-rays in objects with irregular shapes and made it possible to express some expectations regarding other materials, as well. Above all, it is necessary to avoid areas with sharp edges, where the X-ray intensity ratios of individual elements could be significantly affected.

If the surface of an object is curved and it is not possible to focus on even a small, flat area, it is better to choose convex parts of the object for analysis (spheres in presented simulations) than the concave ones (sphere-shaped holes) because the obtained mean concentrations were close to the correct values for convex geometry.

It is recommended that researchers perform more XRF analyses on various points of a certain material. Individual analyses can provide quite different concentrations of an element, but the average concentration may be close to the correct value. This can also partially eliminate the effect of material heterogeneity, which is not discussed in this paper. 

If two elements produce characteristic X-rays with similar energies and with similar X-ray attenuation in a material, then the surface effects increase or decrease their X-ray intensities in the same way. Therefore, the determined concentration ratio for these two elements is nearly correct. A typical example is copper and zinc in copper-based alloys, as well as manganese and iron in steel.

If an element with both K- and L-lines appears in a spectrum (e.g., silver, cadmium, or tin), a technician has to choose one of these X-ray lines for the quantitative analysis. It seems that better results could be obtained with L-lines when the major elements were copper or gold. This statement should also be valid for other major elements with even lower X-ray energies, e.g., iron. Therefore, it seemed that the use of L-lines is usually better for the reduction of surface effect, but one must consider that these low-energy L-lines come from the thin surface layer of a material, and they can be significantly affected by corrosion processes, for instance.

If a desired part of a surface is analyzed with an XRF micro-analysis and an object under investigation cannot be measured with reference geometry, i.e., it is curved or tilted, Table 2 and Figure 14 show that tilting in the axis given by the intersection of the plane of the surface of an object and the plane of the incident and detected beams (x-axis, in this case) did not cause significant disruption in the quantitative analysis.

## Figures and Tables

**Figure 1 materials-16-01133-f001:**
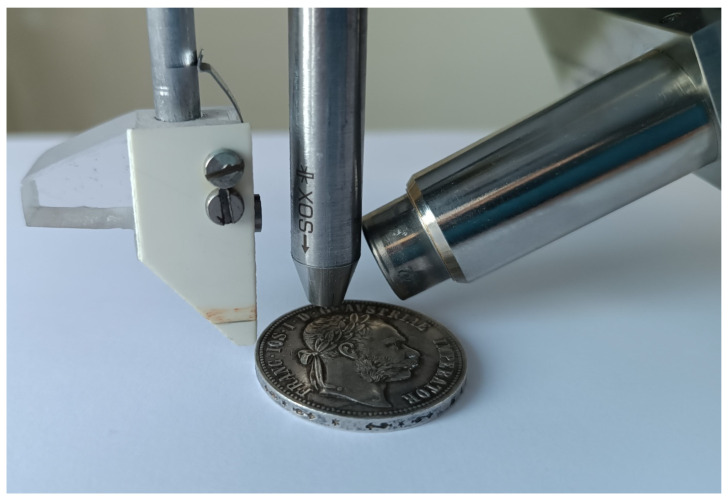
X-ray fluorescence system consisting of SDD detector (**right**), X-ray tube with polycapillary optics device (**middle**), and distance bar (**left**) with a coin as a specimen.

**Figure 2 materials-16-01133-f002:**
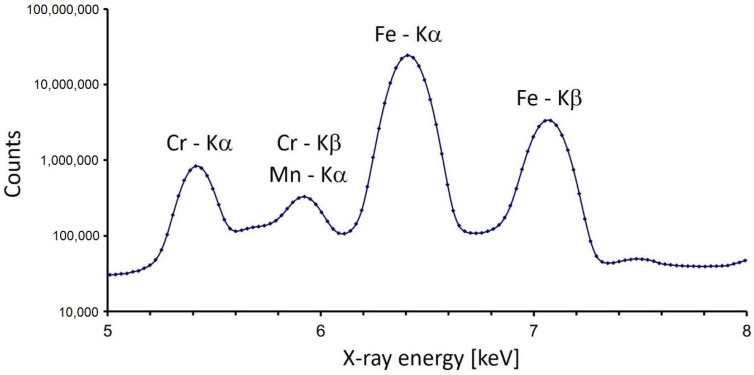
Sum spectrum from the XRF scanning of a modern steel ball bearing.

**Figure 3 materials-16-01133-f003:**
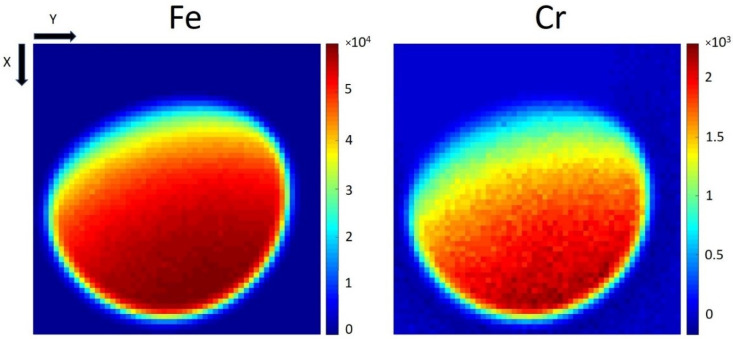
A number of detected photons of Fe and Cr (Kα lines).

**Figure 4 materials-16-01133-f004:**
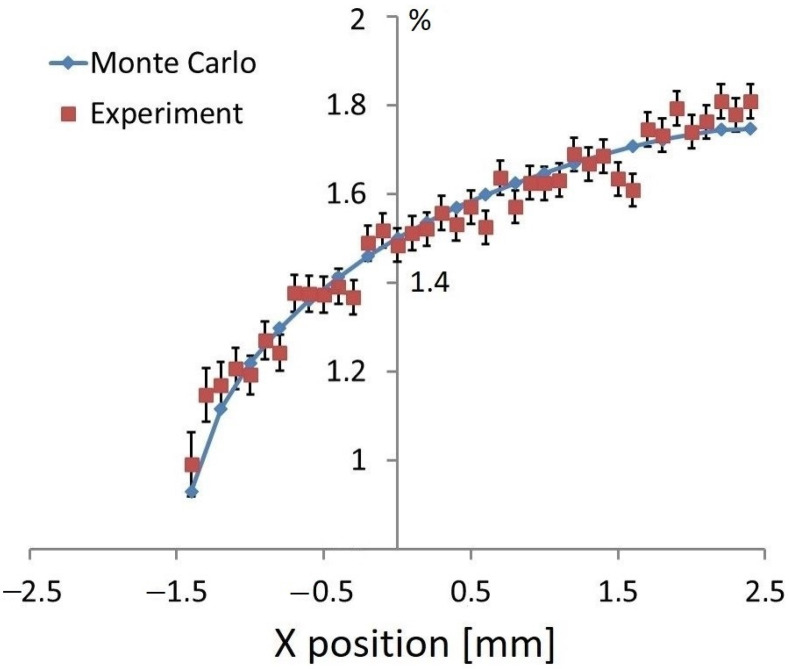
Concentrations of Cr in steel ball bearings were determined with Monte Carlo simulations and experiments. A correct value of 1.5% was obtained for position “0” corresponding to reference geometry (excitation X-ray beam was perpendicular to the surface of the ball).

**Figure 5 materials-16-01133-f005:**
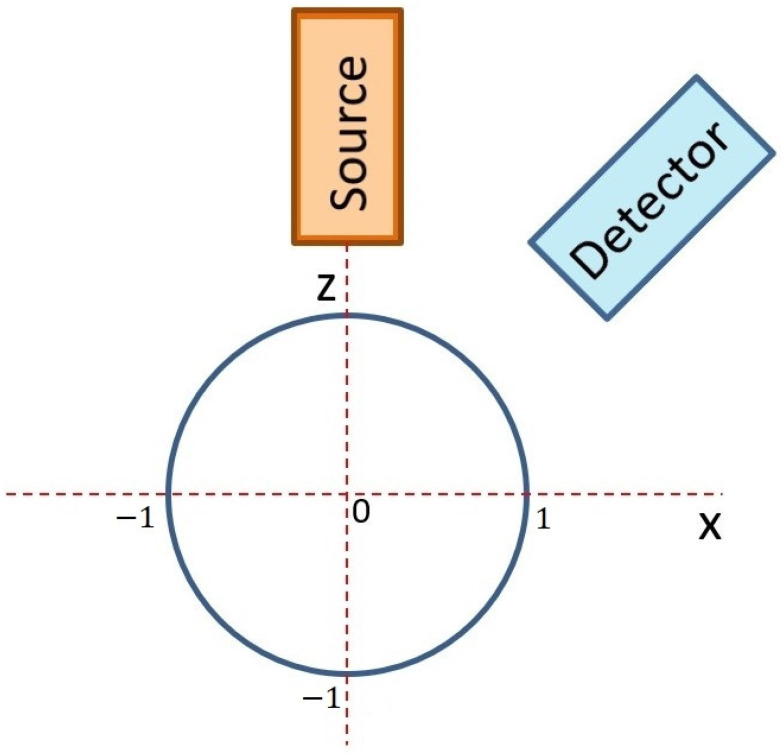
Scheme of Monte Carlo simulation of XRF scanning of an artifact represented by a sphere with a radius of 1 mm.

**Figure 6 materials-16-01133-f006:**
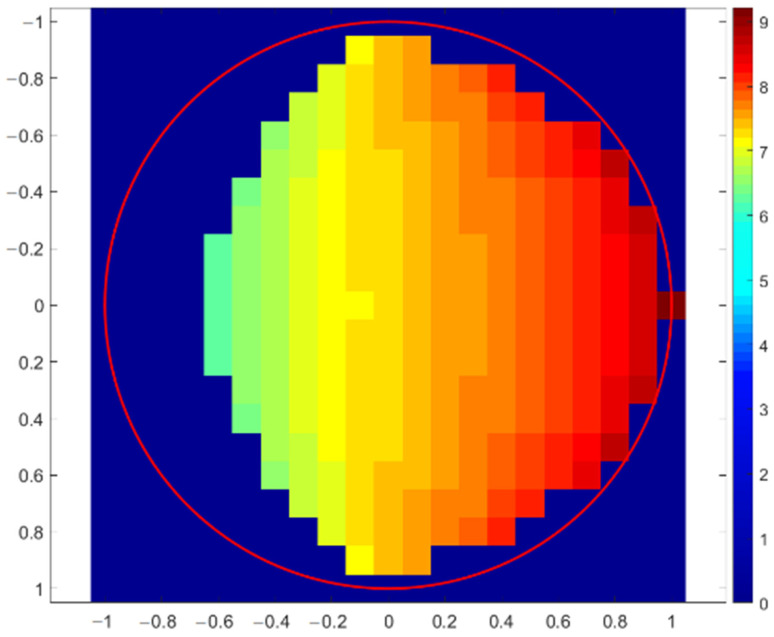
Copper concentration map of sterling silver sphere determined with Monte Carlo simulation of XRF scanning (in XY-axes) applying Cu-Kα and Ag-L lines. The correct Cu concentration was 7.5%.

**Figure 7 materials-16-01133-f007:**
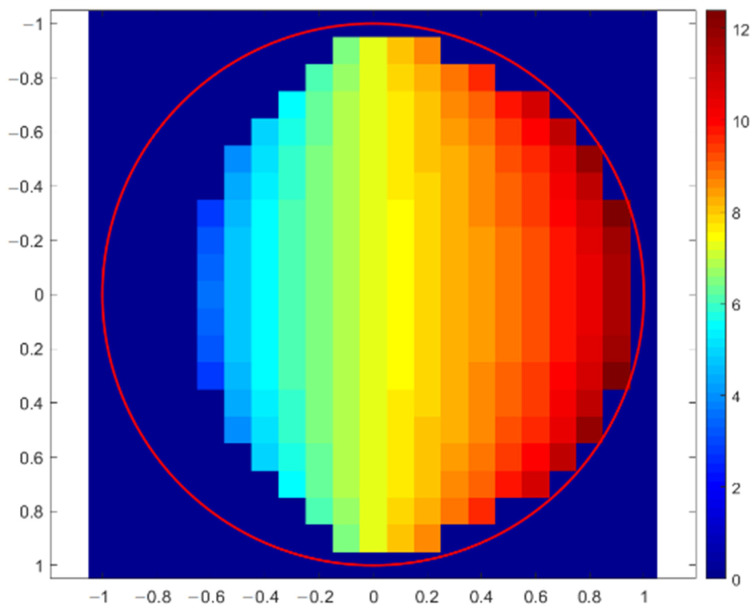
Copper concentration map of sterling silver sphere determined with Monte Carlo simulation of XRF scanning (in XY-axes) applying Cu-Kα and Ag-Kα lines. The correct Cu concentration was 7.5%.

**Figure 8 materials-16-01133-f008:**
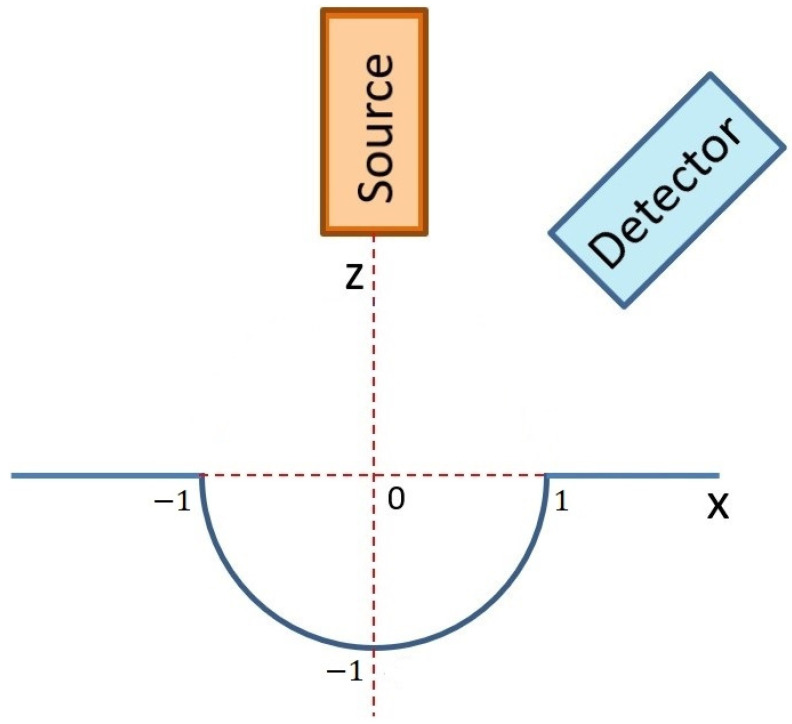
Scheme of Monte Carlo simulation of XRF scanning of an artifact represented by a sphere-shaped hole with a radius of 1 mm.

**Figure 9 materials-16-01133-f009:**
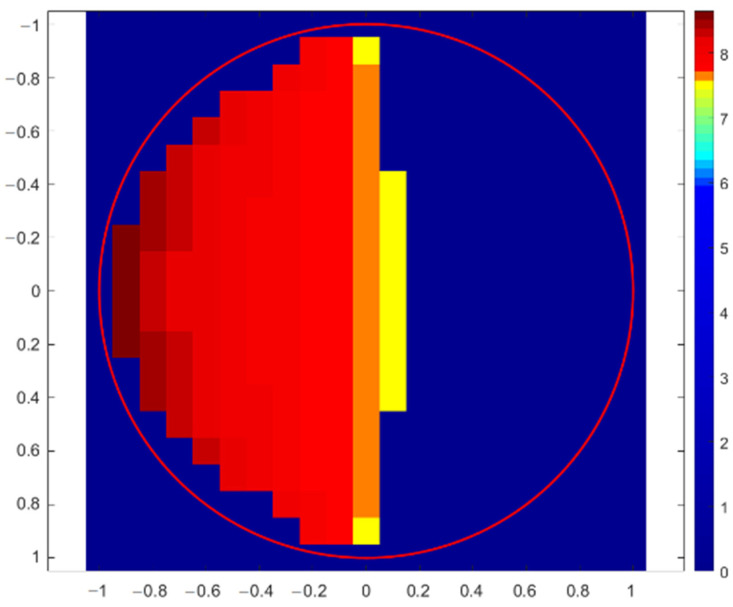
Copper concentration map of sterling silver sphere-shaped hole determined with Monte Carlo simulation of XRF scanning applying Cu-Kα and Ag-L lines. The correct Cu concentration was 7.5%.

**Figure 10 materials-16-01133-f010:**
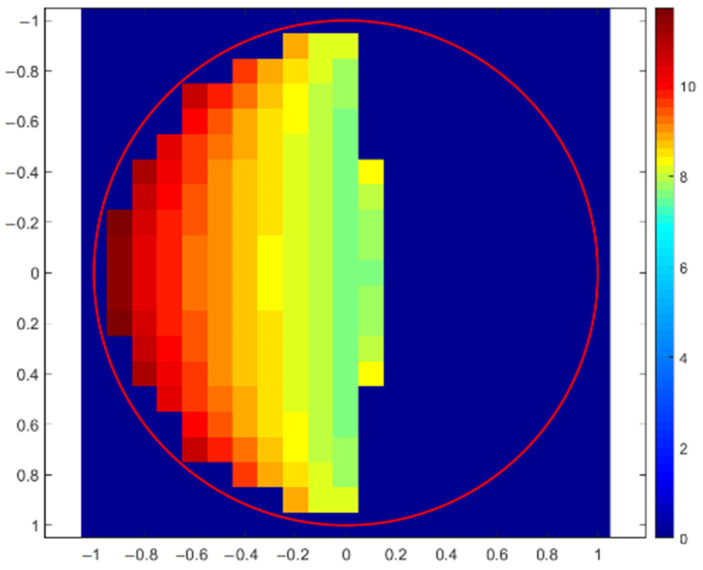
Copper concentration map of sterling silver sphere-shaped hole determined with Monte Carlo simulation of XRF scanning applying Cu-Kα and Ag-Kα lines. The correct Cu concentration was 7.5%.

**Figure 11 materials-16-01133-f011:**
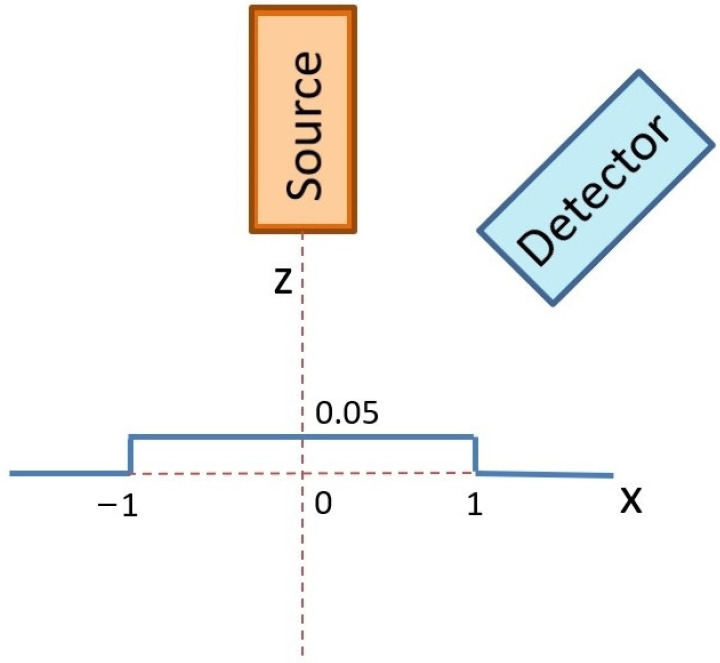
Scheme of Monte Carlo simulation of XRF scanning of an artifact represented by two sharp edges with a height of 0.05 mm.

**Figure 12 materials-16-01133-f012:**
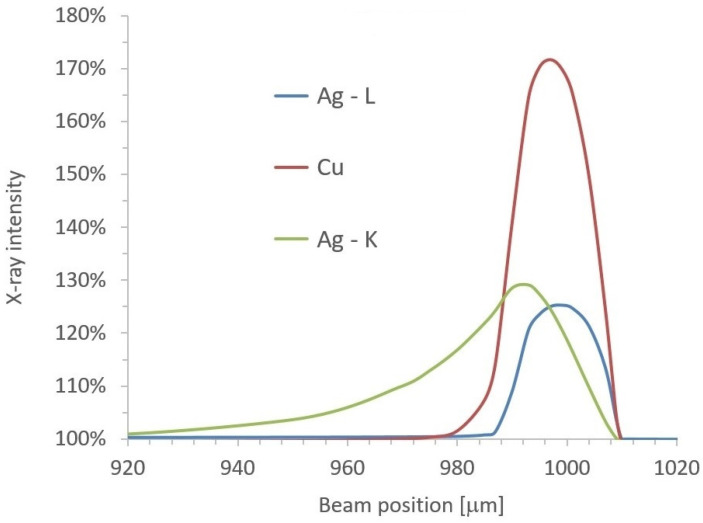
Relative X-ray intensities of copper and silver close to sharp edge on the right (x = 1 mm).

**Figure 13 materials-16-01133-f013:**
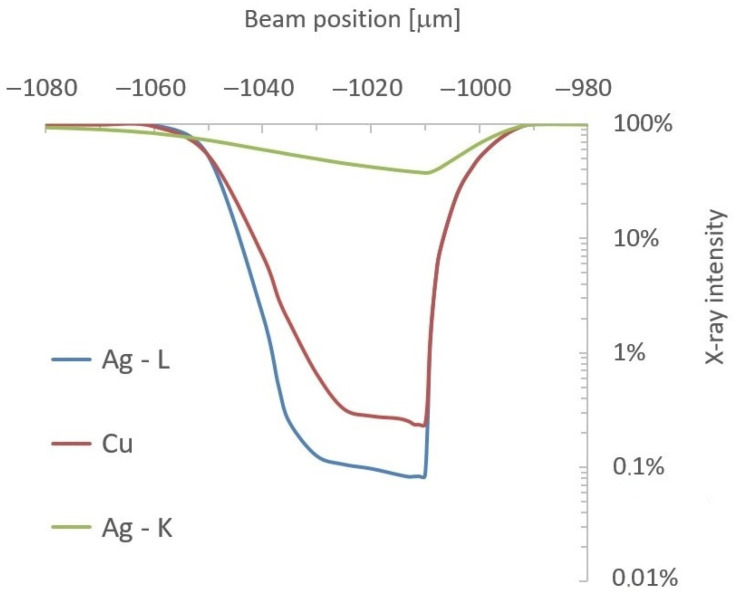
Relative X-ray intensities of copper and silver close to sharp edge on the left (x = −1 mm).

**Figure 14 materials-16-01133-f014:**
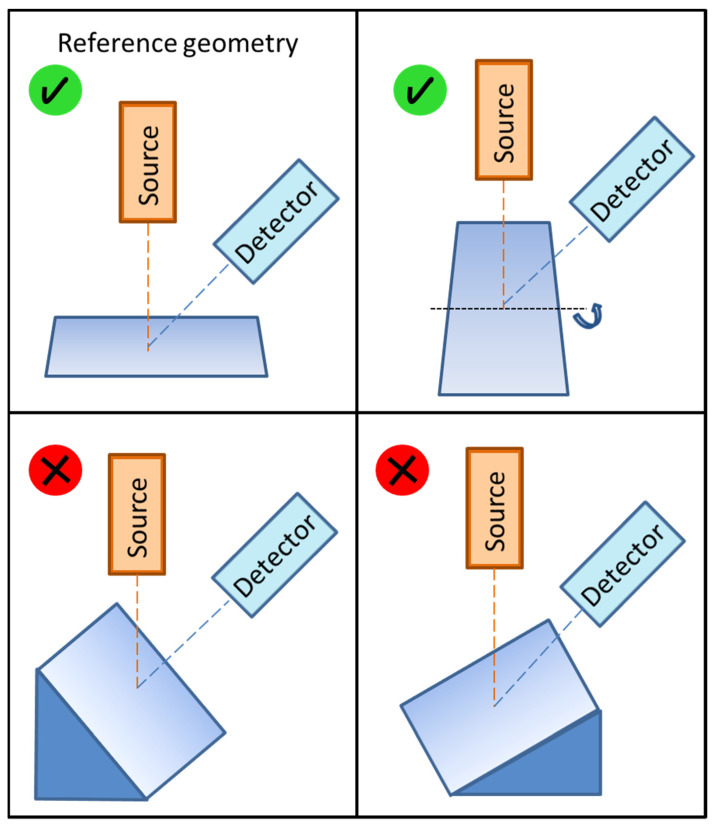
Scheme of the XRF of an object with an indication of the correct analysis (√) in reference geometry and in the case of tilting in the x-axis; wrong analysis (×) was considered for tilting in the y-axis (inclination is shown in the bottom left image).

**Table 1 materials-16-01133-t001:** Summary of Monte Carlo simulations of XRF micro-analyses of various materials with irregular shapes.

Shape and Composition of Materials	Range of Concentrations [%]	Mean Concentration ± Standard Deviation [%]	Standard Deviation/Mean Concentration Ratio
(a) sphere; Ag 92.5, Cu 7.5 (Ag-L used)	Cu: 6.29–8.55	Cu: 7.44 ± 0.63	Cu: 0.085
(b) sphere; Ag 92.5, Cu 7.5 (Ag-Kα used)	Cu: 3.50–11.47	Cu: 7.64 ± 2.09	Cu: 0.281
(c) sphere-shaped hole; Ag 92.5, Cu 7.5 (Ag-L used)	Cu: 7.56–8.60	Cu: 8.01 ± 0.32	Cu: 0.040
(d) sphere-shaped hole; Ag 92.5, Cu 7.5 (Ag-Kα used)	Cu: 7.67–11.40	Cu: 8.94 ± 1.18	Cu: 0.132
(e) sphere; Ag 50, Cu 50 (Ag-L used)	Cu: 48.1–53.1	Cu: 50.0 ± 1.3	Cu: 0.026
(f) sphere; Ag 50, Cu 50 (Ag-Kα used)	Cu: 41.3–53.9	Cu: 49.7 ± 3.1	Cu: 0.062
(g) sphere; Au 75, Ag 12.5, Cu 12.5 (Ag-L and Au-L used)	Au: 70.2–78.8	Au: 74.3 ± 2.3	Au: 0.031
	Ag: 10.2–16.8	Ag: 13.4 ± 1.7	Ag: 0.127
	Cu: 11.0–13.0	Cu: 12.3 ± 0.6	Cu: 0.049
(h) sphere; Cu 79, Sn 7, Pb 14 (Sn-L and Pb-L used)	Cu: 74.3–84.3	Cu: 78.9 ± 2.7	Cu: 0.034
	Sn: 4.68–9.01	Sn: 6.96 ± 1.16	Sn: 0.166
	Pb: 11.1–16.7	Pb: 14.1 ± 1.5	Pb: 0.106
(i) sphere; Fe 95, Mn 5	Mn: 4.25–5.37	Mn: 5.00 ± 0.31	Mn: 0.062

**Table 2 materials-16-01133-t002:** Results of XRF analysis of a piece of copper alloy demonstrating the effect of a tilted surface.

Geometry	Cu [%]	Zn [%]	Sn [%]	Pb [%]
Reference geometry	84.3 ± 0.5	3.39 ± 0.03	5.07 ± 0.39	7.27 ± 0.39
Tilting in x-axis at 30°	84.4 ± 0.8	3.46 ± 0.06	4.85 ± 0.44	7.17 ± 0.52
Tilting in y-axis at 30° (inclination)	80.9 ± 0.3	3.26 ± 0.02	6.53 ± 0.18	9.18 ± 0.39
Tilting in y-axis at 30° (opposite direction)	86.8 ± 0.4	3.58 ± 0.08	3.66 ± 0.21	5.86 ± 0.41

## Data Availability

Not applicable.

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
