# Peer review of "Uncertainty of Quantitative X-ray Fluorescence Micro-Analysis of Metallic Artifacts Caused by Their Curved Shapes"

_materials, 2023, doi:10.3390/ma16031133_

Round 1
Reviewer 2 Report
Dear Editors and Authors,
The article entitled “Uncertainty of quantitative X-ray fluorescence micro-analysis of metallic artefacts caused by their curved shapes" is interesting and well-prepared. This research investigates the effects of an irregular shape on the result of a quantitative X- 10 ray fluorescence (XRF) micro-analysis. The authors evaluate the accuracy and precision of the XRF by performing several measurements and computer simulations. At the end, some recommendations are given. After carefully reading the article, I would accept this article for publication in this journal after the authors doing some minor revisions:
1. Abstract: Please mention the important finding and significance of this research in the conclusion part
2. Introduction: The aims of the study shall be explicitly mentioned.
3. Materials and Method: Please divide the section into several sub-section: Equipment, materials/samples, methods, etc.
4. Conclusion: The title can be improved to Conclusions and future perspective / recommendation. Please make the recommendation in a paragraph, not point per point.
Reviewer 3 Report
The manuscript “Uncertainty of quantitative X-ray fluorescence micro-analysis of metallic artefacts caused by their curved shapes” deals with the effects of irregular specimen shape in the analysis of quantitative XRF for metallic samples (mainly Cu, Ag and Au alloys). The authors performed measurements and complementary simulations to evaluate the accuracy of the XRF.
Comments to authors:
Keywords: it seems to be missing “fluorescence” in the keyword “X-ray”
Recommend review of the introduction section
Line 26. “gamma rays in this particular case”- you are using X-rays in this particular case, not gamma rays (line 79: “air-cooled X-ray tube with a Mo anode”). I recommend to modify this part of the sentence to “X-rays in this particular case” or to “ X-rays or gamma rays”
“Since the energy of characteristic X-rays exactly corresponds to an atomic number, individual chemical elements can be identified and even quantified, because the intensity of characteristic X-rays is related to a concentration of an element in a material.” – recommend to re-write this sentence: the energy of characteristic X-rays exactly corresponds to the energy difference between the excited and the ground state of the inner shell electrons, and this value is particular for each shell transition and element.
Line 41: change recognized by identified
Line 51: It is presumed that unknown samples and reference materials are analyzed under the same experimental conditions and have similar matrix.
2. Materials and Methods
Line 81: has attached a polycapillary… According to the manufacturer of the polycapillary
Parameters of the tabletop XRF system for micro-analysis are described in detail in the article [14]. -> this paper is not open access. Describe the main information in this section.
Monte Carlo code MCNP: name the code (is MCNP “Monte Carlo N-Particle Transport”?)
Line 99: i.e. Tally F5 – energy fluence spectrum in the position of the detector was scored. -> didn’t understand what it means.
The computing time was several minutes for an individual task. -> be more precise. Takes n minutes for x task.
Figure 1: better description if the figure. X-ray fluorescence system consisting of xyz with a coin as specimen.
3. Results and discussion
3.1: suggest to add an image of the ball, or change Fig1 for one with the ball
Line 114, 192, 199, 270, 296: “was hit with the X-ray micro-beam” -> suggestion to change “hit” to “illuminated by the X-ray micro-beam / in which the X-ray beam was focused”
118: acquisition time was only 2 s per one spot
Fig 3: A Number of detected photons of Fe and Cr (Kα lines)
Line 150: values of detected photons
Line 159 imitating -> fitting
Figures 5, 8, 11: change the color of the axis or the object. The 2 blues are too close and impacts the visualization of the object. The resolution of the images can be improved.
Line 270: (position x = 1 mm = 1000 m) -> the micron symbol is missing.
Line 273: The question was whether the increase in X-ray intensities would be the same or similar for all these 3 X-ray lines
Line 317: made a lot of Monte -> include the number of simulations instead of “a lot”
Line 341: I recommend to also include this information in the first time you mention to use Ag-L and gets better results than using Ag-Ka, so the unfamiliar reader can already understand the reason for this decision.
“Since the attenuation coefficients for Cu-Ka and Ag-L lines are closer to each other, the irregular shape of the surface increases or decreases these two lines quite similarly, i.e., the intensity ratio of these two lines remains almost constant, and thus the element concentrations are close to correct values if normalization of concentrations (to 100%) is considered.”
Line 376 and Fig 14 and table 2: inform which of the images in figure 14 correspond to which part of the text and in the table.
With respect to figure 14, indicate the angles between sample-source, sample-detector and source-detector.
Table 2: in the columns of Cu and Zn [%], there is “Cu:” in front of the numbers.
Why the values of Cu are in a range and the value of Zn, Sn and Pb are an average?
